# Green synthesis of graphene oxide by seconds timescale water electrolytic oxidation

Songfeng Pei[1], Qinwei Wei[1,2], Kun Huang[1], Hui-Ming Cheng [1,3] & Wencai Ren [1]

Graphene oxide is highly desired for printing electronics, catalysis, energy storage, separation membranes, biomedicine, and composites. However, the present synthesis methods depend on the reactions of graphite with mixed strong oxidants, which suffer from explosion risk, serious environmental pollution, and long-reaction time up to hundreds of hours. Here, we report a scalable, safe and green method to synthesize graphene oxide with a high yield based on water electrolytic oxidation of graphite. The graphite lattice is fully oxidized within a few seconds in our electrochemical oxidation reaction, and the graphene oxide obtained is similar to those achieved by the present methods. We also discuss the synthesis mechanism and demonstrate continuous and controlled synthesis of graphene oxide and its use for transparent conductive films, strong papers, and ultra-light elastic aerogels.

[1] Shenyang National Laboratory for Materials Science, Institute of Metal Research, Chinese Academy of Sciences, 72 Wenhua Road, 110016 Shenyang, China. [2] School of Materials Science and Engineering, University of Science and Technology of China, 72 Wenhua Road, 110016 Shenyang, China. [3] Tsinghua-Berkeley Shenzhen Institute (TBSI), Tsinghua University, 1001 Xueyuan Road, 518055 Shenzhen, China. Songfeng Pei and Qinwei Wei contributed equally to this work. Correspondence and requests for materials should be addressed to W.R. (email: wcren@imr.ac.cn)

Graphene oxide (GO), which were usually used as a precursor for the synthesis of graphene, but is attracting increasing interest because of many unique properties and promising applications that are different from graphene[1–4]. In GO, a great number of oxygen-containing functional groups such as hydroxyl, epoxy, and carboxyl are covalently bonded to graphene basal plane and edges with a C/O atomic ratio of around 2.0[1–3], which makes GO sheets strongly hydrophilic and can be well dispersed in water[1–6]. The stable and homogeneous GO suspensions allow easy assembly of various graphene macrostructures with tunable fascinating properties such as transparent conductive films[7, 8], strong and stiff papers[9], multifunctional separation membranes[10, 11], conductive and strong fibers[12] and ultralight super-elastic aerogels[13, 14]. Moreover, the oxygen-containing functional groups enable GO to be easily functionalized and strongly interacted with other materials, which endures GO a chemically tunable platform and a wide range of technological applications such as supercapacitors and lithium batteries[15, 16], fuel cells[17], biomedicine[18], and polymer composites[19, 20].

Generally, GO is synthesized by exfoliation of graphite oxide[1–21]. Since the first preparation by Brodie in 1859[22], the Brodie and the subsequently developed Staudenmaier[23] and Hummers[24] methods have been the three primary routes for synthesizing graphite oxide, which all depend on the homogeneous reactions of graphite with mixed strong oxidants (Supplementary Table 1). Both Brodie and Staudenmaier methods use $KClO_3$ and nitric acid (most commonly fuming [higher than 90% purity]), and suffers from explosion risk, as well as production of hazardous gases (e.g., $NO_x$ and $ClO_2$) and carcinogenic $ClO^-$. The most commonly used Hummers methods[1–21, 24, 25] use a large amount of concentrated $H_2SO_4$ and $KMnO_4$ to ensure sufficient oxidation. The highly reactive $Mn_2O_7$ intermediates can cause severe explosion at elevated temperatures as well. Inevitably, all these three methods produce serious environmental pollution and metal ion impurities on GO sheets. For instance, in Hummers method, around 1000 times more water than graphite has to be used to remove the excessive $H_2SO_4$ and $KMnO_4$ after oxidation reaction (Supplementary Table 1), producing a huge amount of waste water containing mixed acids and heavy metal ions, and $Mn^{2+}$ are usually detected on GO sheets. Moreover, they are all time-consuming, which need a few to hundreds of hours for oxidation (Supplementary Table 1). Although the oxidation time can be reduced to about 1 h by using strong oxidant $K_2FeO_4$ and concentrated $H_2SO_4$[26], more oxidizing mixtures are used (Supplementary Table 1), leading to more pollution.

Recently, electrochemical (EC) processes have been developed and widely used to synthesize graphene sheets because of the environmental-friendliness, high efficiency and relatively low cost[27] (Supplementary Table 2). It has been shown that nearly pristine graphene with a C/O of 25.3 and mobility of $405 cm^2 \cdot V^{-1} s^{-1}$ could be synthesized within seconds by EC exfoliation of graphite foil[28]. Inspired by these successes, researchers also tried to synthesize GO by EC oxidation of various graphitic materials such as pencile core[29], graphite rode[30] and graphite flakes[31, 32]. Unfortunately, the accompanied water/solvent electrolysis process aggravates the expansion and delamination of graphitic materials, which lead to ineffective current supply or broken circuit before the desired EC oxidation process can be achieved[27, 28]. Therefore, the products suffer from low oxidation and exfoliation degree[29–35] (Supplementary Table 3), which are much different from the GO synthesized by traditional chemical oxidation methods mentioned above. Although the products synthesized by EC oxidation of graphite rode show a low C/O atomic ratio of 2.2, they are mainly oxidized at edges and

electrically conductive[30]. Recently, an EC Tee-cell setup has been used to try to ensure the electrical current to all graphitic flakes, however, the product shows a maximum C/O atomic ratio of 3.0 even after about 48 h reaction[34].

Here, we report a scalable, safe, ultrafast, and green method to synthesize clean GO sheets by water electrolytic oxidation of graphite. It is found that the pre-intercalation of graphite efficiently inhibit the anodic electrocatalytic oxygen evolution reaction of water at high voltage to enable the ultrafast oxidation of graphene lattice within a few seconds, which is over 100 times faster than the present methods. The GO obtained shows similar chemical composition, structure, and properties to those achieved by traditional Hummers method. As examples, we demonstrate its use for high-performance transparent conductive films, strong papers and ultra-light elastic aerogels. Moreover, this method enables continuous production and easy control on the oxidation degree, number of layers and lateral size of GO sheets.

## Results

**Synthesis and characterization of GO sheets by EC method.** The synthesis of GO contains two sequential EC processes at room temperature with commercial flexible graphite paper (FGP) as raw material (Fig. 1a and Supplementary Figs. 1, 2). The commercial FGP has similar structure with graphite, high tensile strength (4 to 5 MPa), excellent electrical conductivity comparable with HOPG, and good flexibility (Fig. 1b, Supplementary Note 1, and Supplementary Figs. 3, 4). It usually has a thickness from micrometers to millimeters, width up to meters, and length up to kilometers. First, the FGP is subjected to EC intercalation in concentrated $H_2SO_4$ to form stage-I graphite intercalation compound paper[36] (GICP, Fig. 1c and Supplementary Movie 1). Then, the GICP is used as anode for EC reaction in diluted $H_2SO_4$ (50 wt.%). Very surprisingly, the blue-colored GICP dipped in diluted $H_2SO_4$ is oxidized quickly to yellow-colored graphite oxide[24] within a few seconds along with exfoliation (Fig. 1d, Supplementary Movies 2, 3, and Supplementary Fig. 5). After vacuum filtration and cleaning with water, the filter cake is exfoliated in water by sonication to form electrochemically synthesized GO (EGO) dispersion (Fig. 1e).

It is worth noting that the total amount of water used for cleaning graphite oxide in our method (mass ratio to raw FGP, around 150, Supplementary Note 2 and Supplementary Fig. 6) is significantly lower than those used in the Hummers methods (typically around 1000). Similar to the GO synthesized by traditional Hummers method (HGO)[5], our EGO shows highly stable solubility in water. It retains a homogeneous dispersion, without any precipitates and color change, after storage for 6 months at a concentration of $1 mg \cdot mL^{-1}$ in water (Fig. 2a). After freezing drying, the yield of EGO is estimated to be around 96 wt.% of raw FGP. In addition, this method is very easy to scale up. EGO can be continuously produced by continuously introducing the GICP slice into dilute $H_2SO_4$ solution with a specific rate (Supplementary Movie 2). For example, using two 250 mL beakers as reactors and a 0.5 mm-thick, 40 mm-wide FGP as raw material, EGO was continuously produced with a productivity of about $12 g \cdot h^{-1}$, showing the great potential of this method for mass production of GO sheets. It is needed to point out that FGP is an ideal raw material for continuous production of GO sheets by our method because of its combined excellent properties mentioned above and the good tolerance to the volume expansion caused by intercalation shown below, although it is relatively difficult being oxidized by traditional Hummers method comparing with natural graphite flakes (Supplementary Note 1 and Supplementary Table 4). Graphite powders cannot be used directly in our method because of the

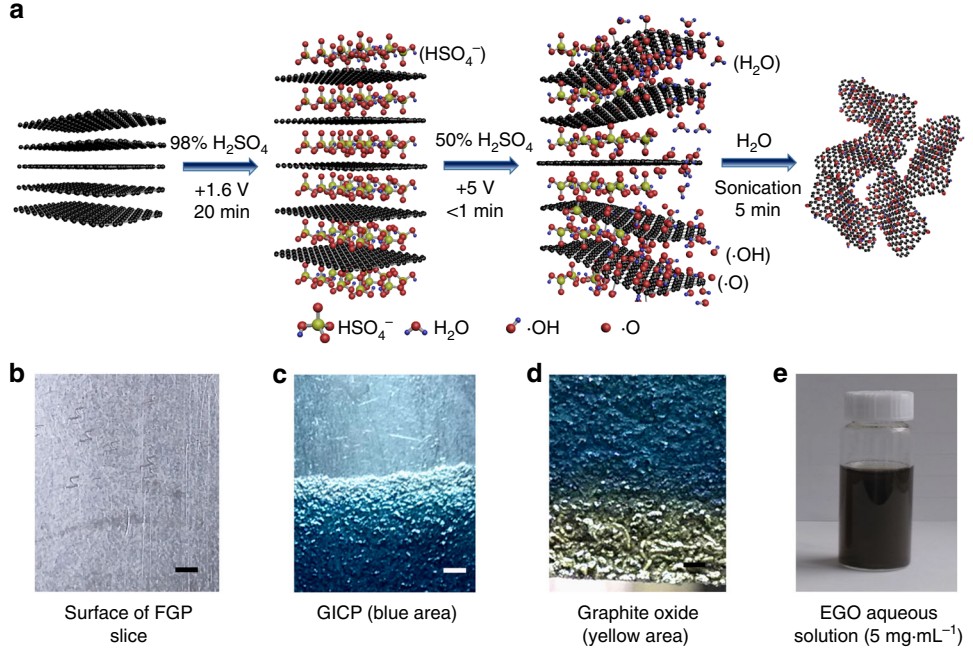

**Fig. 1** Synthesis of EGO by water electrolytic oxidation. **a**, Schematic illustration of the synthesis process of EGO by water electrolytic oxidation. **b–e**, Photos of the raw material and the products obtained at each step. **b**, FGP. **c**, GICP (blue area) obtained after EC intercalation of FGP in 98 wt.% $H_2SO_4$ at 1.6 V for 20 min. **d**, Graphite oxide (yellow area) obtained by water electrolytic oxidation of the GICP in 50 wt.% $H_2SO_4$ at 5 V for 30 s. **e**, Well-dispersed EGO aqueous solution (5 mg·mL$^{-1}$) obtained by sonication of the graphite oxide in water for 5 min. Scale bars in **b-d**: 1 mm

very small size and have to be fabricated into macroscopic electrode to ensure voltage supply. Graphite rods are also not suitable for our method since they are easily broken into small pieces even after short time intercalation and cannot be used for further oxidation.

The chemical composition of the continuously produced EGO was analyzed by combustion elemental analyzer (EA), X-ray photoelectron spectrometer (XPS), Fourier transform infrared spectrometer (FTIR), and inductively coupled plasma atomic emission spectrometer (ICP-AES). The EA analysis shows that EGO has a typical composition (at. %) of C (50.2), O (29.2), H (19.7), and S (0.9). This gives C/O atomic ratio of around 1.7, which is lower than the typical value for HGO (about 2.0)[7, 24, 26], indicating a higher oxidation degree of EGO. XPS C1s spectra show strong C=C peak (284.6 eV), prominent epoxy/hydroxyl peak (C–O, 287.0 eV), and weak carbonyl (C=O, 288.0 eV) and carboxyl (O–C=O, 289.2 eV) peaks (Fig. 2b). FTIR spectra show O–H stretching vibration (in the range of 3000 to 3700 cm$^{-1}$), C=O vibration (1740 cm$^{-1}$), C=C vibration from $sp^2$ bonds (1645 cm$^{-1}$) and C–O vibration (1402, 1087, and 1042 cm$^{-1}$) (Fig. 2c). These XPS and FTIR features are similar to those of HGO, suggesting that EGO has the same oxygen-containing functional groups as HGO. The abundant functional groups enables EGO a high zeta potential of −56.8 mV (pH = 6.8), leading to excellent solubility in water, as shown in Fig. 2a. Consistent with the fact that no metal-containing chemicals are used, ICP-AES measurements confirm the absence (less than 50 ppb) of typical metal impurities (Fe, Co, Ni, Mn, Cu, Zn) in EGO.

It is well accepted that GO has a structure of small $sp^2$ C–C domains isolated within the $sp^3$ C–O matrix[1–3]. Raman spectroscopy provides a powerful tool to obtain the detailed structural information of graphene-based materials. The G peak is attributed to bond stretching of $sp^2$ carbon pairs, and the D peak is attributed to the breathing mode of $sp^2$ carbon rings and activated by defects[37]. As shown in Fig. 2d, both EGO and HGO show prominent D peak (1323 cm$^{-1}$) and G peak (1580 cm$^{-1}$),

and very weak 2D peak (2650 cm$^{-1}$) and D + G peak (2903 cm$^{-1}$). It is worth noting that the intensity ratio of D peak to G peak in EGO is higher than that in HGO, suggesting that the $sp^2$ domains in EGO are smaller than those in HGO. This is consistent with the higher oxidation degree of EGO shown above.

We then used an atomic force microscope (AFM), scanning electron microscope (SEM), high resolution transmission electron microscope (HRTEM) to characterize the number of layers and lateral size of EGO (Fig. 2e–j). Similar to HGO, the monolayer EGO sheets show a thickness of about 1.0 nm (Fig. 2e) because of the presence of functional groups and adsorbed water. AFM and SEM measurements indicate that the lateral size of EGO sheets is mostly in the range of 1 μm to 10 μm (about 61%), with about 7% larger than 10 μm and about 32% smaller than 1 μm (Fig. 2i). HRTEM images of the folded edges give direct evidence on the number of layers (Fig. 2h). Extensive HRTEM observations on 150 EGO sheets show that about 46% of them are monolayer and about 86% are no more than three layers (Fig. 2j), which are consistent with the AFM measurement results (Supplementary Fig. 7). Figure 2k shows the mass-produced EGO aqueous solution (5 mg·mL$^{-1}$). Note that there are also some multilayers in the GO samples exfoliated from the graphite oxide made by Hummers method without centrifugation[25, 38]. In our method, extending the sonication time greatly reduces the percentage of multilayers and the lateral size of the EGO sheets (Fig. 2i, j and Supplementary Fig. 7). For instance, the samples obtained by 30 min sonication contain about 95% monolayers and about 5% bilayers, and the lateral size is mostly smaller than 1 μm.

**Synthesis mechanism and controlled synthesis of GO sheets by EC method.** To understand the synthesis mechanism of EGO, we used in-situ Raman spectroscopy (Fig. 3a, e), ex-situ XRD (Fig. 3b, f) and XPS (Fig. 3c, g), and optical microscopy (Fig. 3d) to monitor the structure and chemical composition changes of a FGP slice with reaction time in the two-step EC process. As

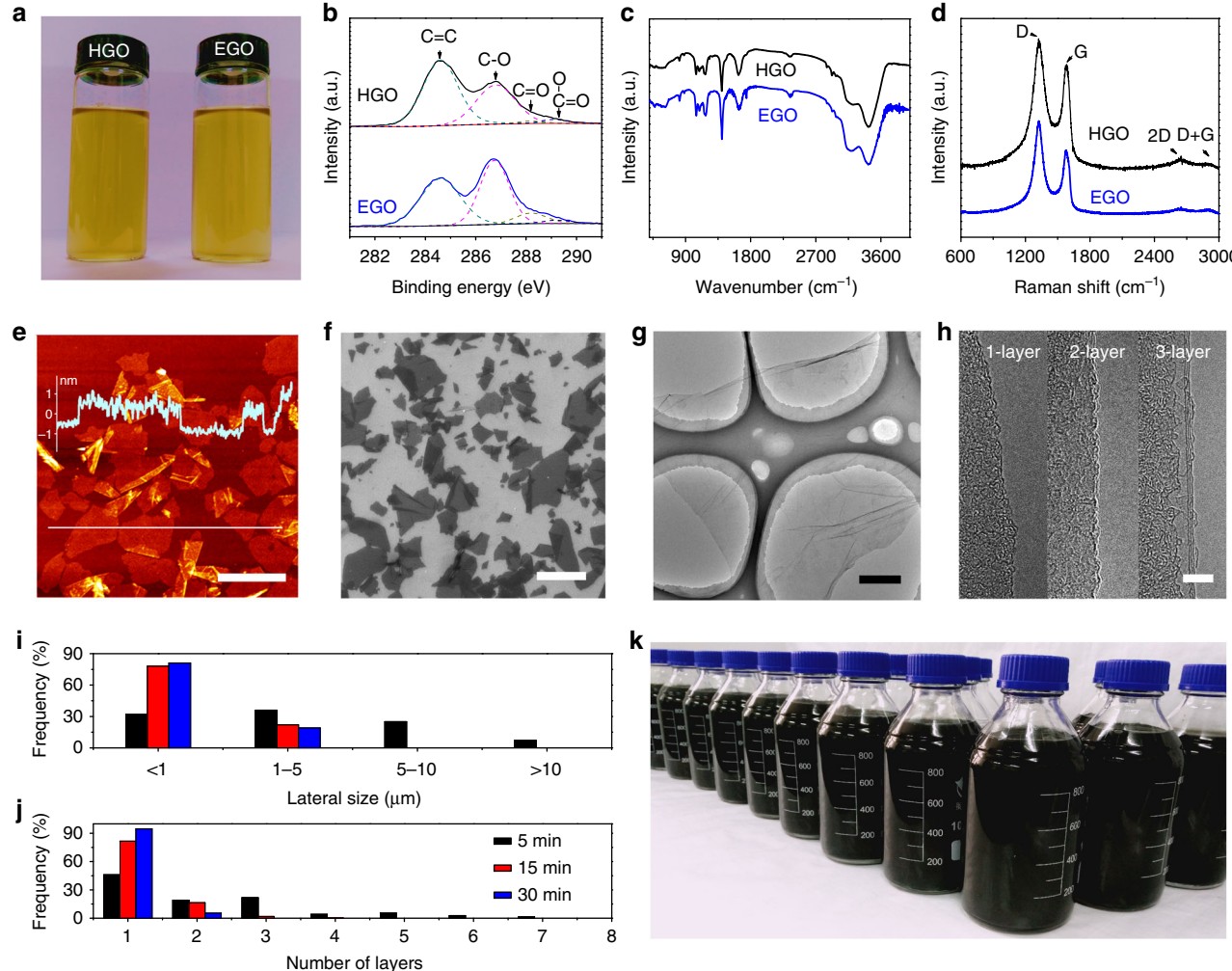

**Fig. 2** Characterization of EGO. **a-d**, Comparison of EGO with HGO: aqueous solution (**a**, 1 mg·mL$^{-1}$), XPS C1s spectra (**b**), FTIR spectra (**c**), Raman spectra (**d**). HGO was synthesized by a modified Hummers method[7] using graphite flakes (80 mesh) as raw material. **e–h**, Typical AFM (**e**), SEM (**f**), TEM (**g**), and HRTEM (**h**) images of EGO sheets. **i, j**, Lateral size (**i**), and number of layers (**j**) distribution of EGO sheets obtained by sonication for 5 (black columns), 15 (red columns), and 30 min (blue columns). **k**, Mass produced EGO aqueous solution (5 mg·mL$^{-1}$). Scale bars: **e**, 2 μm; **f**, 10 μm; **g**, 1 μm; **h**, 5 nm

shown in Figs. 1b, 3a and 3b, after 20 min reaction in concentrated H$_2$SO$_4$, the FGP slice changes color from lustering-grey to blue, the Raman G peak shifts from 1580 to 1631 cm$^{-1}$ and the XRD (002) peak shifts from 26.5° to 23.2°, which give clear evidence for stage-I intercalation of graphite[36, 39]. It is worth noting that there are no defects or oxidation-containing functional groups in the obtained GICP (Fig. 3a, c, Supplementary Note 3, and Supplementary Fig. 8). Moreover, the GICP remains nearly the same fracture force and flexibility with FGP, and the surface resistance greatly decreases because of the doping of sulfuric acid (Supplementary Table 5). Although no oxidation occurs after a long-reaction time (2 h) in concentrated H$_2$SO$_4$, the GICP suffers severe swelling (about 7 times increase in thickness compared to FGP, Supplementary Table 5), which leads to dramatic decrease in mechanical strength and conductivity, and therefore cannot be used in the following EC process.

To show the structure and composition change of GICP clearly during the second EC process, we fixed a GICP slice in the electrolyte (initial dipping length: 1 cm) without moving. During EC process, once the graphite was oxidized, it became insulating, and consequently its EC oxidation reaction stopped immediately although it was still kept in the electrolyte. Very surprisingly, as shown in Fig. 3d, Supplementary Movie 3 and Supplementary Fig. 5, only after a few seconds, some areas of the GICP surface

change from blue to yellow, evidence of the oxidation of graphite[24]. With extending the reaction time, the yellow areas quickly expand and the whole surface becomes yellow after about 60 s. Then, the color becomes darker and darker with the oxidation of the interior part, and no color changes any more after 3 min. The evolutions of Raman spectra, XRD pattern and XPS C1s spectra confirm the full oxidation of GICP from the surface to the interior (Fig. 3e–g). During the continuous production process of EGO, a GICP slice was continuously introduced into the electrolyte at a speed of around 5 mm·min$^{-1}$ but the GICP dipped in the electrolyte remained almost a constant length of around 1 mm, as shown in Supplementary Movie 2. All these results suggest that the EC oxidation of graphite lattice finishes within a few seconds. JM Tour and co-workers studied the formation mechanism of graphene oxide by a modified Hummers method using a single graphite flake, and found that the conversion of the graphite intercalation compound into graphite oxide, a key step in the formation of GO, takes a few hours even days[40]. Therefore, the EC oxidation rate in our method is over 100 times faster than the oxidation rate of the Hummers methods.

Another very important feature of our method is that the oxidation degree of the graphite oxide obtained can be easily tuned by simply changing the concentration of H$_2$SO$_4$ in

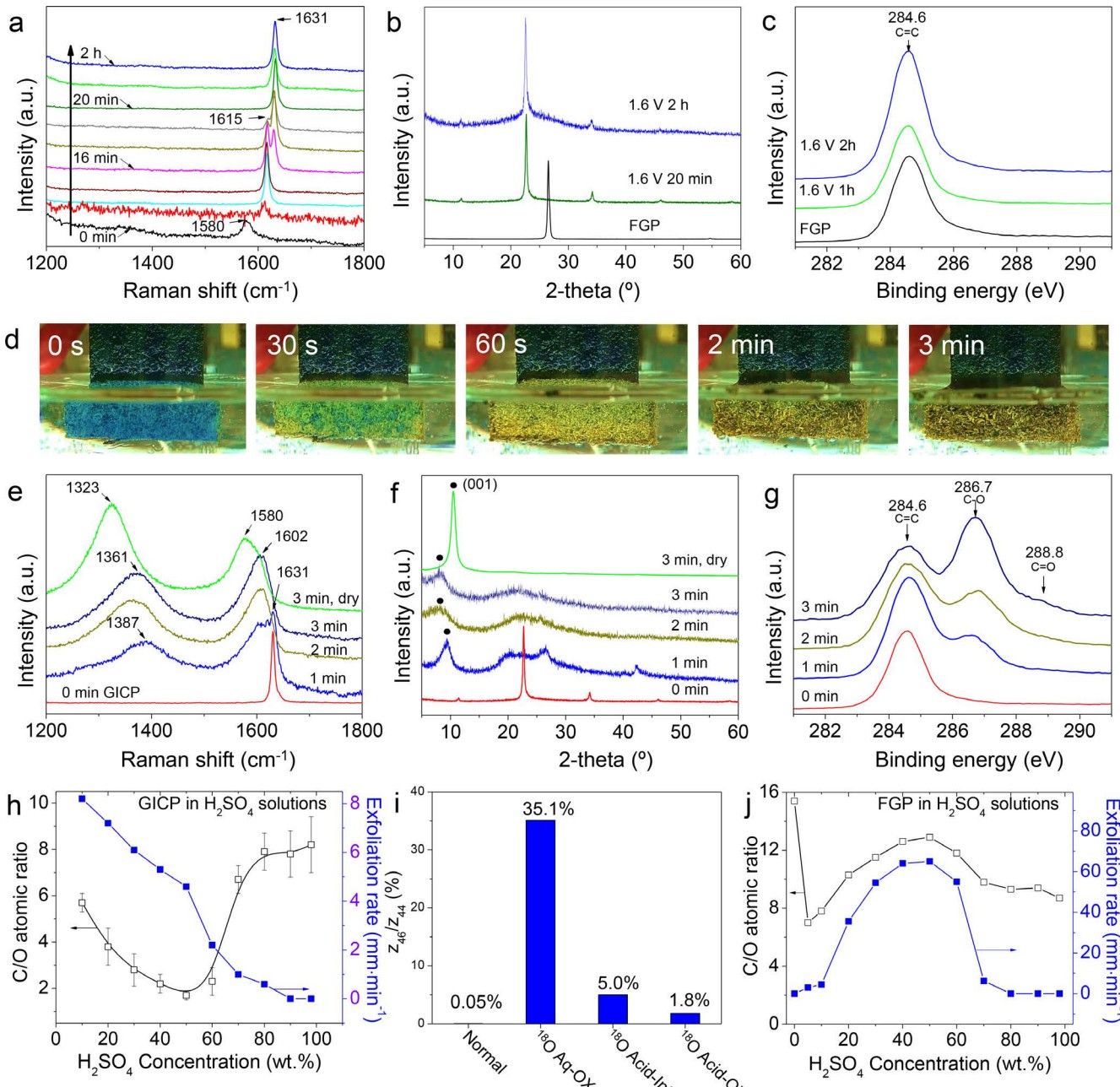

**Fig. 3** Investigation on the EC synthesis process of EGO. **a–c**, Step I: EC intercalation. Raman spectra (**a**), XRD patterns (**b**), and XPS C1s spectra (**c**) after intercalation for different time. From bottom to top in (**a**), the reaction time is 0, 1, 4, 10, 16, 17, 18, 20, 60, and 120 min in sequence. **d–g**, Step II: EC oxidation. **d**, Morphology and color change of a GICP slice with reaction time. Raman spectra (**e**), XRD patterns (**f**), and XPS C1s spectra (**g**) after reaction for different time. The Raman spectra and XRD patterns indicated by 0, 1, 2, and 3 min in (**e**) and (**f**) were taken from the as-synthesized GICP and EGO samples after 1, 2, and 3 min reaction, respectively, without cleaning and drying, and those indicated by 3 min, dry were taken from the samples obtained after 3 min reaction followed by cleaning and drying. It can be found that the contained water and $H_2SO_4$ have significant influences on the Raman spectra and XRD patterns (also see Supplementary Note 4 and Supplementary Fig. 9). The XPS C1s spectra in (**g**) were taken from the samples after cleaning and drying. **h**, C/O atomic ratio of the exfoliated product and the corresponding exfoliation rate of GICP in $H_2SO_4$ solutions with different concentrations. The error bar shows the range of C/O atomic ratio measured from five samples in each batch of product. **i**, Comparison of $^{18}O$ content in the EGO samples synthesized by four different combinations of the normal and $^{18}O$-containing reagents, including $H_2SO_4$ for EC intercalation (normal Acid-Int/$^{18}O$ Acid-Int), and water (normal Aq-OX/$^{18}O$ Aq-OX) and $H_2SO_4$ (normal Acid-OX/$^{18}O$ Acid-OX) used in EC oxidation process. Normal: EGO synthesized with normal Acid-Int, Aq-OX, and Acid-OX; $^{18}O$ Aq-OX: EGO synthesized with normal Acid-Int, $^{18}O$ Aq-OX, and normal Acid-OX; $^{18}O$ Acid-Int: EGO synthesized with $^{18}O$ Acid-Int, normal Aq-OX, and normal Acid-OX; $^{18}O$ Acid-OX: EGO synthesized with normal Acid-Int, normal Aq-OX, and $^{18}O$ Acid-OX. (**j**), C/O atomic ratio of the exfoliated product and the corresponding exfoliation rate of FGP in $H_2SO_4$ solutions with different concentrations

electrolyte, as shown in Fig. 3h (also see Supplementary Note 5, Supplementary Fig. 10, and Supplementary Table 6). Highly oxidized GO (C/O < 2) is achieved only when the $H_2SO_4$ concentration is in the range of 40 wt.% to 60 wt.%, and the highest oxidation degree with C/O atomic ratio of 1.5−1.8 is achieved at a $H_2SO_4$ concentration of 50 wt.%. Below and above this concentration range, only partially oxidized products (C/O > 2) are obtained. Moreover, mass spectrometry analysis of the gaseous products of the EC oxidation process shows no sulfur-containing byproduct (Supplementary Note 6, Supplementary Fig. 11a–d, and Supplementary Table 7), indicating that $H_2SO_4$ is not decomposed. These results suggest that the water in electrolyte plays a key role in the EC synthesis of graphite oxide. Further [18]O isotropic tracing experiments clearly reveal that the oxygen functional groups in EGO dominantly originate from water (Fig. 3i, Supplementary Note 7 and Supplementary Fig. 12).

We also studied the EC reaction of FGP directly in dilute $H_2SO_4$ aqueous solution. However, only weakly oxidized graphene sheets with a C/O atomic ratio larger than 7 or thin graphite flakes were obtained (Fig. 3j, Supplementary Fig. 10 and Supplementary Table 6). This is similar to the reported EC methods[27, 28, 34], which directly use pristine graphitic materials for EC reaction in acid or salt aqueous solution and can only produce pristine or weakly oxidized graphene sheets with a C/O atomic ratio mostly larger than 3.0 even after a very long reaction time of 48 h (Supplementary Tables 2 and 3). Therefore, the pre-intercalation of graphite by $H_2SO_4$ plays a key role in ultrafast EC synthesis of highly oxidized GO sheets in our method.

It is well established that the anodic electrocatalytic oxygen evolution reaction of water occurs under an applied voltage, and it contains four elementary reactions as following[41]:

$$* + H_2O \overset{E}{\Rightarrow} *OH + H^+ + e^- \qquad (1)$$

$$*OH \overset{E}{\Rightarrow} *O + H^+ + e^- \qquad (2)$$

$$*O + H_2O \overset{E}{\Rightarrow} *OOH + H^+ + e^- \qquad (3)$$

$$*OOH \overset{E}{\Rightarrow} O_2 \uparrow + H^+ + e^- \qquad (4)$$

Where * stands for an active site on electrode surface, *OH, *O, and *OOH represent the radical intermediates adsorbed on the active site, and E refers to the driving voltage. In the case of graphitic anode, the adsorbed reactive *OH, *O, and *OOH can react with the carbon lattice that is highly positively charged to form covalently bonded oxygen-containing functional groups[34], [42]. However, the rapid formation of a great number of $O_2$ gas will exacerbate the exfoliation of the graphitic anode, leading to ineffective current supply or broken circuit and consequently stopping the electrochemical oxidation reaction[27, 28].

We then studied the gaseous products of the EC reactions with GICP, FGP, or Pt as anode and Pt as cathode in 50 wt.% $H_2SO_4$ aqueous solution at 5 V (Supplementary Note 6 and Supplementary Fig. 11e). It was found that the mole ratio of $O_2$ to $H_2$ is only 1:8.36 for GICP as anode, which is significantly smaller than that produced by water electrolysis with Pt as both anode and cathode (1:2.05). In contrast, the mole ratio of $O_2$ to $H_2$ is about 1:3.4 for FGP as anode. As a result, the exfoliation rate of GICP is about one order of magnitude slower than that of FGP (Fig. 3h, j and Supplementary Movies 3, 4). These results indicate that the use of GICP efficiently inhibits the formation of $O_2$ at a high voltage.

The sufficient adsorbed reactive *OH, *O, and *OOH together with the high voltage enable the ultrafast synthesis of fully oxidized graphite oxide and GO (Fig. 3d–g). For FGP as anode, because of the absence of sufficient reactive *OH, *O, and *OOH and exfoliation induced broken circuit, only graphene sheets or graphite flakes with a low oxidation degree are obtained (Fig. 3j). It has been reported that 2,2,6,6-tetramethylpiperidin-1-oxyl (TEMPO) can efficiently suppress the formation of oxygen radicals from water electrolysis[28]. When adding 4‰ and 10‰ of TEMPO in 50 wt.% $H_2SO_4$ aqueous solution during the optimized EC oxidation process of GICP, the C/O ratio of the products increases from 1.8 to 2.2 and 2.6, respectively (Supplementary Note 8, Supplementary Fig. 13 and Supplementary Table 8). This gives evidence of the existing of oxygen radicals and their key role in the synthesis of GO during our EC oxidation process.

## Discussion

The above results suggest that EGO is synthesized by water electrolytic oxidation of graphite, which is conceptually different from the present methods. In this EC method, $H_2SO_4$ mainly acts as a control agent to tune the anodic electrocatalytic oxygen evolution reaction of water to enable the ultrafast oxidation of graphene lattice, and no other oxidants are used. Therefore, there is no explosive risk and metal ion contaminations, and $H_2SO_4$ can be fully recycled (Supplementary Note 9). Second, the EC oxidation rate is over 100 times faster than the oxidation rate of Hummers and $K_2FeO_4$ methods. Third, the obtained graphite oxide is mixed with dilute $H_2SO_4$ solution (50 wt.%). In the Hummers method, however, the graphite oxide is homogeneously mixed with concentrated $H_2SO_4$ and other oxidants, forming very viscous slurry. Therefore, the cleaning of graphite oxide in our method is much easier, which needs ten times less of water. Fourth, this method has a good controllability. The oxidation degree of GO sheets can be easily tuned by changing the concentration of $H_2SO_4$ solution in the EC oxidation process, and the number of layers and lateral size can be tuned by sonication time. Fifth, continuous production of GO sheets can be realized easily by continuously introducing GICP into dilute $H_2SO_4$, as shown in Supplementary Movie 2. Therefore, our EC oxidation method combines the advantages of safety, ultrafast synthesis, easy control, environmentally friendliness, no metal ion contaminations and easiness to scale up, which paves the way for industrial production and applications of GO sheets at a low cost.

As examples, we demonstrated the use of EGO for transparent conductive films (TCFs) (Fig. 4a–c), flexible and strong papers (Fig. 4d–f) and ultra-light elastic aerogels (Fig. 4g–i), which are typical applications of HGO. After reduction by hydroiodic (HI) acid, the EGO-based TCF shows a surface resistance of about 1.5 kΩ·□$^{-1}$ with a transparency of 80% at 550 nm, which is comparable to those of the TCFs made by HGO with a similar lateral size[7]. The EGO paper shows a well-aligned layered structure and a mechanical strength of 175 MPa, which is stronger than the papers made by HGO (120 MPa)[9] and $K_2FeO_4$-GO with larger lateral size (140 MPa)[26]. The EGO aerogel has a highly porous structure with a density of about 3 mg·cm$^{-3}$, but is highly elastic. As shown in Fig. 4i, it can sustain their structural integrity under a load of more than 1000 times their own weight and can rapidly recover from more than 80% compression.

## Methods

**Materials**. FGP and natural graphite flakes (80 mesh, mean particle size of about 150 μm) were purchased from Shenyang RuiYu Chemical Co. Ltd, and fully dried in an oven at 80 °C for 24 h before use. Concentrated sulfuric acid (98 wt.%, $H_2SO_4$), phosphorus pentoxide (Analytical reagent [AR], $P_2O_5$), ethylenediamine (AR, $C_2H_8N_2$), potassium permanganate (AR, $KMnO_4$), and sodium nitrate (AR,

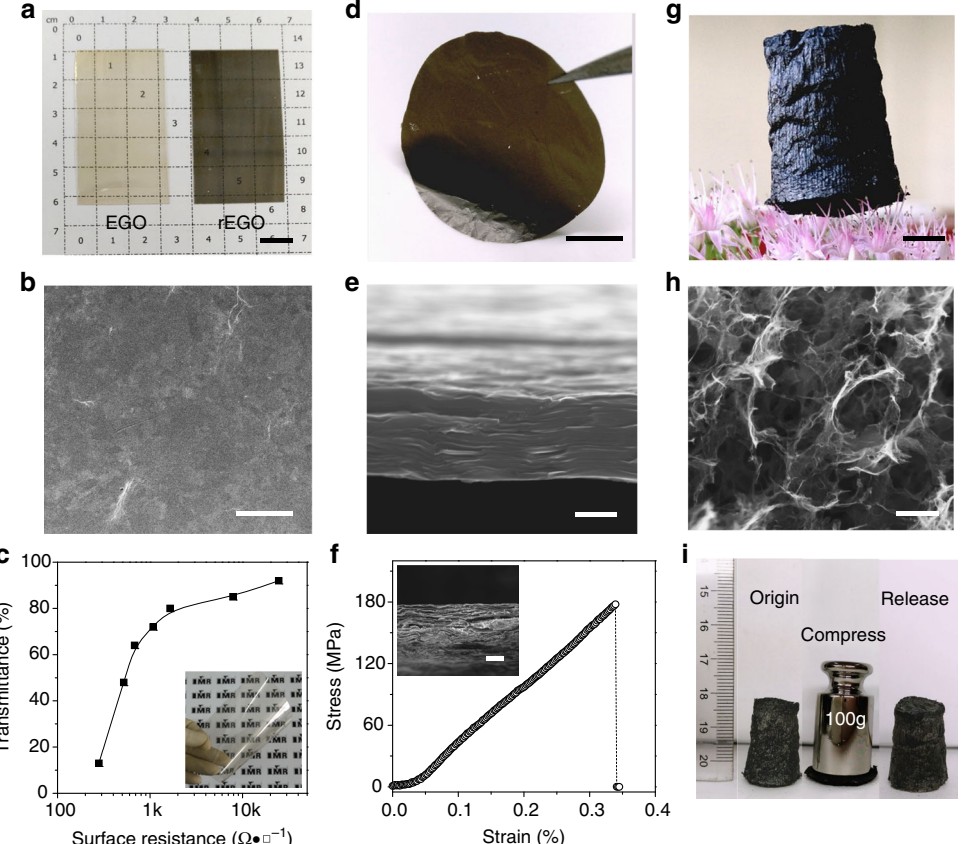

**Fig. 4** Macrostructures assembled from EGO. **a–c**, TCF. **a**, Photos of as-prepared (left) and reduced (right) ultra-thin EGO films on PET substrates. **b**, SEM image of the surface of reduced ultra-thin EGO film. **c**, Transmittance vs surface resistance of the reduced ultra-thin EGO films (inset). **d–f**, Flexible and strong paper. **d**, Photo. **e**, Side-view SEM image. **f**, Tensile curve of a 30 μm-thick paper, and the inset shows the fracture plane. **g–i**, Ultralight and elastic aerogel. **g**, An aerogel standing on pistil of a flower. **h**, SEM image. **i**, Elastic property of a aerogel with a density of 3 mg·cm$^{-3}$. Scale bars: **a**, **d**, **g**, 1 cm; **b**, **f**, **h**, 10 μm; **e**, 2 μm

NaNO$_3$) were purchased from Sinopharm Chemical Reagent Co., Ltd. and used as received. Distilled water was used as solvent and reagent. Platinum (Pt) wire with a diameter of 0.4 mm (99.99%, Shenyang RuiYu Chemical Co. Ltd.) was used as cathode in EC intercalation and oxidation processes. Heavy-oxygen water (97 atom %, H$_2^{18}$O) and TEMPO (98%) were purchased from Shanghai Aladdin Bio-chemical Technology Co., Ltd. and used as received. SO$_3$ gas was first synthesized by the reaction of 98 wt.% H$_2$SO$_4$ and excessive P$_2$O$_5$, and then excessive SO$_3$ was absorbed by H$_2^{18}$O to form the concentrated $^{18}$O-containing sulfuric acid (H$_2$S$^{16}$O$_3^{18}$O).

**EGO synthesis**. The fully dried FGP slice was first cut into slices with a dimension of 10 × 4 cm$^2$, and then dipped into a beaker (250 mL) containing 200 mL concentrated H$_2$SO$_4$ (98 wt.%) for EC intercalation, in which the FGP slice was used as anode and Pt wire as cathode with a DC power supply (Bio Logic, VSP-300 multichannel electrochemical workstation) of 1.6 V. After 20 min reaction, the GICP obtained was taken out and pressed to remove the absorbed H$_2$SO$_4$. Then the GICP was subjected to EC oxidation. In this process, the GICP was used as anode and Pt wire as cathode with a DC power supply of 5 V, and a diluted H$_2$SO$_4$ solution (50 wt.%) worked as the electrolyte. In order to keep a balance between the dipping and exfoliation, the GICP slice gradually dipped into a beaker (250 mL) containing 200 mL electrolyte at a speed of 5 mm·min$^{-1}$ controlled by a pro-grammed automated lifting platform. The exfoliated graphite oxide was collected by vacuum filtration. The filter cake obtained was washed several times with dis-tilled water to remove absorbed acid, and then dispersed in water by short time sonication to form EGO solution. All the unproductive H$_2$SO$_4$ involved in the synthesis process of EGO, including the absorbed concentrated H$_2$SO$_4$ on GICP and the very dilute H$_2$SO$_4$ solution obtained after distilled water washing, were collected and recycled (Supplementary Note 9).

**Characterization of EGO**. The microstructure was characterized by SEM (FEI Nova NanoSEM 430), HRTEM (FEI Tecnai G2 20), AFM (Bruker, Dimension FastScan with ScanAsyst$^{TM}$, operating in the tapping mode), and Raman spec-troscopy (JY Labram HR 800,632.8 nm laser). The chemical composition was

evaluated by a combustion elemental analyzer (Elementar, vario MICRO cube), XPS (ESCALAB 250 using Al Kα radiation source), FTIR (Bruker Tensor 27), and ICP-AES (IRIS Intrepid, THERMO ELEMENTAL). XPS spectra were fitted using the XPS peak 4.1 software in which a Shirley background was assumed. For comparison, the Raman, XPS and FTIR spectra of HGO sheets were also measured using the same instruments and same conditions. Zeta-potential was measured with Malvern Zetasizer Nano-ZS90.

**In-situ Raman spectroscopy and ex-situ XRD and XPS measurements**. The structure and composition evolution of FGP with reaction time in the two EC processes were monitored by in-situ Raman and ex-situ XRD and XPS measure-ments. The Raman spectra were measured in-situ with a home-made sample holder. The integral time was 40 s, and 20 s was used for re-focusing of laser beam on the expanded sample surface. For XRD measurements, the samples were first filtered to remove liquid phase without washing, and then measured on D/Max-2400 with Cu Kα radiation ($k = 1.5418$ Å). For XPS measurements, the samples were collected, washed and exfoliated with water to remove absorbed sulfuric acid and form suspensions. Then a small amount of suspension was subjected to vacuum filtration, and finally the thin film obtained was dried at 40 °C for 24 h for XPS measurements.

**Isotopic tracing experiments**. The oxygen-containing reagents used for EGO synthesis can be divided into three types: concentrated H$_2$SO$_4$ used for EC inter-calation (Acid-Int), water (Aq-OX), and H$_2$SO$_4$ (Acid-OX) in electrolyte for EC oxidation. To trace the transfer path of oxygen, we replaced these three reagents by their isotopic homologs (H$_2^{18}$O or H$_2$S$^{16}$O$_3^{18}$O) for EGO synthesis. Four different EGO samples were synthesized with different combinations of reagents: Normal sample, which was synthesized with normal reagents with no $^{18}$O addition; $^{18}$O Aq-OX sample, which was synthesized with normal H$_2$SO$_4$ for intercalation but electrolyte containing H$_2^{18}$O (97 atom%) and normal H$_2$SO$_4$ for oxidation; $^{18}$O Acid-Int sample, which was synthesized with H$_2$S$^{16}$O$_3^{18}$O (98 wt.%) for inter-calation and normal reagents for oxidation; $^{18}$O Acid-OX sample, which was synthesized with normal H$_2$SO$_4$ for intercalation but electrolyte containing

$H_2S^{16}O_3^{18}O$ and normal $H_2O$ for oxidation. After freezing drying, the $^{18}O$ content in each EGO sample was measured by thermogravimetric mass spectrometry (TG-MS) with Netzsch STA 449C Jupiter/QMS 403C.

**Fabrication and measurements of GO macrostructures**. A TCF was fabricated by rod-coating of EGO aqueous solution on a PET substrate followed by chemical reduction. Typically, EGO aqueous solution (0.5 mg·mL⁻¹) was first coated with a Mayer rod (type: 150 μm) on the surface of a PET substrate, which had been treated to be hydrophilic by short time oxygen plasma treatment. The resulting wet coating was then dried at 50 °C for 2 h to form an EGO film on the PET substrate. Then the EGO coating was reduced by immersing into the hot HI acid (40 wt.%, 80 °C) and held for 15 min. After reduction, the PET slice with reduced EGO coating was washed in ethanol and water in sequence, and finally blow dried with nitrogen gas followed by thermal treatment at 80 °C for 1 h. The coating thickness was controlled by changing the concentration of EGO solution used for rod-coating. The transparency of the TCF (transmittance at 550 nm wavelength) was measured by a UV-vis-NIR spectrometer (Varain Cary 5000) with a pure PET substrate as reference. The surface resistance of the TCF was measured by a 4-point probe resistivity measurement system (RTS-9, Guangzhou, China).

EGO paper was fabricated by vacuum filtration of the solution containing mildly reduced EGO sheets. Typically, 0.5 g of 40 wt.% HI acid was first added into 100 mL of EGO aqueous solution (0.5 mg·mL⁻¹), then the solution was heated to 60 °C and mildly stirred for 30 min. During this process, the solution gradually changed color from pale-yellow to black, but was still well dispersed without obvious aggregation. After that, the dispersion was subjected to vacuum filtration for about 30 min to obtain a film on the surface of a filter membrane (CMC membrane, pore size 0.45 μm). After drying at 80 °C for 30 min, the film was peeled off from the filter membrane surface to get a free-standing EGO paper. For tensile strength measurements, the EGO paper was cut into narrow strips with a dimension of $10 \times 2$ mm², and measured with a Hounsfield H5K-S materials tester. The thickness of the strip was measured with a spiral micrometer.

EGO aerogel was fabricated by hydrothermal treatment followed by freezing drying. Typically, 20 μL of ethylenediamine was first mixed into 5 mL EGO aqueous solution (3 mg·mL⁻¹). Then the solution was heated for 6 h at 95 °C in a sealed glass vial to form a hydrogel. After freezing drying, the obtained aerogel was heated at 150 °C overnight to fully eliminate the absorbed water. The compressive test was performed manually with a weight of 100 g.

**Data availability**. The data that support the findings of this study are available from the corresponding author upon request.

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

## Acknowledgements

We thank Libo Gao and Long Chen for AFM measurements, Zhibo Liu for TEM measurements, Ying Li, Guodong Wen, Lichang Yin, Chenghua Sun, and Chao Zhen for helpful discussions. This work was financially supported by the National Key R&D Program of China (No. 2016YFA0200101), National Science Foundation of China (Nos. 51325205, 51290273, and 51521091), Chinese Academy of Sciences (Nos. KGZD-EW-

303-1, KGZD-EW-T06, 174321KYSB20160011, and XDPB06), and the Youth Innovation Promotion Association of Chinese Academy of Sciences. All data are in the main paper and supplementary materials.

## Author contributions

W.R. conceived and supervised the project; S.P., Q.W., and W.R. designed the experiments; S.P., Q.W., and K.H. performed the experiments; S.P. and W.R. analyzed the data; S.P., H.C., and W.R. wrote the manuscript. All the authors discussed the results and commented on the manuscript.

## Additional information

**Competing interests:** The authors declare no competing financial interests.

