## [Peer Review File · Nature Communications]

Reviewers' comments:

Reviewer #1 (Remarks to the Author):

This paper reports a two-step electrochemistry method for the preparation of graphene oxide with H₂SO₄ as the only chemical reagent. The graphite can be fully oxidized in a few seconds with a yield of 96 wt.%, and the exfoliated GO sheets show similar chemical composition, microstructure and properties to those achieved by the present methods. Although this paper presents some exciting results, there are some puny issues that need to be further addressed. Thus, I am pleased to recommend this paper be considered after a minor revision according to the below comments.

1. This paper reports an electrochemistry method, but no electrochemistry methods about GO preparation were introduced in the section of Introduction. Thus, more electrochemistry methods need to be listed and compared with the method in this work.
2. The graphene oxide in the paper has a low C/O ratio of 1.7, corresponding to a high degree of oxidation. But why so many multi-layer GOs exist in the final product?
3. The author said that the graphite was fully oxidized within a few seconds. But why the XRD pattern in Fig. 3f shows an obvious graphite peak after 3 mins' oxidation? Please provide more data to prove that the oxidation process finished within a few second.
4. The anodic electrocatalytic oxygen evolution reaction of water occurs under an applied voltage, and it contains four elementary reactions. More data are needed to support the existing of *OH, *O and *OOH.

Reviewer #2 (Remarks to the Author):

This manuscript reports a fast and green method to prepare graphene oxide by electrochemical oxidation of flexible graphite paper in aqueous H₂SO₄ solution. This is an interesting work, and the results are important. I recommend accept this paper for publication after minor revisions.

1. The electrochemical synthesis of graphene oxide has been studied by several groups. The typical papers are listed as follows. (a) Carbon, 2016, 100, 540-545. Synthesis of graphite oxide by electrochemical oxidation in aqueous perchloric acid; (b) J. Mater. Chem. A, 2014, 2, 15428-15436. Synthesis of graphene oxide nanosheets by electrochemical exfoliation of graphite in cetyltrimethylammonium bromide and its application for oxygen reduction; (c) Chem. Commun., 2016, 52, 5714. One-step electrochemical synthesis of nitrogen and sulfur co-doped, high-quality graphene oxide. The authors should compare their work with the previous papers to clarify the novelty and importance of this submission.
2. The authors described: "However, the present synthesis methods depend on the reactions of graphite with strong oxidizing mixtures containing one or more concentrated acids,....., which is 2 -4 orders of magnitude faster than the present methods". However, the comparison of the new method with Hummers methods is based on different raw materials. The authors used flexible graphite paper (FGP) as raw material rather than natural graphite as used in Hummers method. Thus, the above comparison is unfair.
3. The structure and properties of FGP should be studied more clearly. What are the difference between FGP and graphite rod or powders?

Response to Reviewers' Comments

Reviewer #1 (Remarks to the Author):

This paper reports a two-step electrochemistry method for the preparation of graphene oxide with H₂SO₄ as the only chemical reagent. The graphite can be fully oxidized in a few seconds with a yield of 96 wt.%, and the exfoliated GO sheets show similar chemical composition, microstructure and properties to those achieved by the present methods. Although this paper presents some exciting results, there are some puny issues that need to be further addressed. Thus, I am pleased to recommend this paper be considered after a minor revision according to the below comments.

Reply: We thank the reviewer very much for positive comments.

1. This paper reports an electrochemistry method, but no electrochemistry methods about GO preparation were introduced in the section of Introduction. Thus, more electrochemistry methods need to be listed and compared with the method in this work.

Reply: We thank the reviewer very much for kind suggestion.

We have added the discussion on the reported electrochemical methods about GO synthesis in the Introduction section in the revised manuscript. The detailed comparison of our method with the reported electrochemical methods was also given in Supplementary Table 2 and 3.

2. The graphene oxide in the paper has a low C/O ratio of 1.7, corresponding to a high degree of oxidation. But why so many multi-layer GOs exist in the final product?

Reply: We thank the reviewer very much for the comment.

The GO sheets obtained by our electrochemical method indeed have high degree of oxidation. Actually, the GO sheets synthesized from the fully oxidized graphite oxide made by traditional Hummers method also contains many multi-layer GO flakes (*ACS Nano* 4, 4806, 2010; *Carbon* 47, 493, 2009) if no centrifugation was used.

In our method, the presence of multi-layer GO sheets is attributed to the short sonication time (5 min). We have studied the influence of sonication time on the thickness distribution of EGO sheets in the product. It was found that extending the sonication time greatly reduces the percentage of multi-layer EGO sheets (Supplementary Fig. 7). For instance, the samples obtained by 30 min sonication contain ~95% monolayers and ~5% bilayers (Fig. 2j).

We have added the above information in the revised manuscript.

3. The author said that the graphite was fully oxidized within a few seconds. But why the XRD pattern in Fig. 3f shows an obvious graphite peak after 3 mins' oxidation? Please provide more data to prove that the oxidation process finished within a few second.

Reply: We thank the review very much for kind comment and suggestion.

The original XRD patterns and Raman spectra shown in Fig. 3e, f were taken from the as-synthesized samples without cleaning and drying, which contain some water and H₂SO₄. As the reviewer mentioned, they are indeed different from those of the commonly reported fully oxidized GO sheets, which were measured after water cleaning and drying. To understand these differences, we cleaned the as-synthesized samples by water washing followed by drying and measured again. It can be seen that the dried clean samples show the same Raman and XRD characteristics with the fully oxidized GO sheets (the green lines in Figs. 3e, f in the revised manuscript). These results indicates that our samples oxidized for 3 min are really fully oxidized, and the Raman and XRD differences are attributed to the influence of water and H₂SO₄ contained in the as-synthesized samples. It is needed to point out that the XPS spectra shown in Fig. 3g were taken from the samples after cleaning and drying since the as-synthesized samples cannot be measured by XPS.

To clearly show the structure and composition change of GICP during the second EC process, we fixed a GICP slice in the electrolyte (initial dipping length: ~1 cm) without moving. Different from the traditional chemical oxidation methods, the EC

oxidation reaction must be driven by applied voltage. Once the graphite was oxidized, it became insulating, and consequently its EC oxidation reaction stopped immediately even though it was still kept in the electrolyte. As shown in Fig. 3d, Supplementary Fig. 5 and Supplementary movie 3, some areas of the GICP surface changed from blue to yellow only after a few seconds, which gives a strong evidence that the EC oxidation process finished in a few seconds.

XPS is a direct method to characterize the oxidation of graphite. We have tried to characterize the yellow-colored areas that were formed within 10 seconds by using XPS but it is very difficult. For XPS measurements, the absorbed H_2SO_4 and water in the samples must be removed by cleaning and drying. Unfortunately, after these treatments, the oxidized yellow-colored areas inevitably mixed with the un-oxidized areas, and could not be distinguished any more.

In our practical production process, the GICP slice was continuously introduced into the electrolyte at a speed of ~ 5 mm/min controlled by a programmed automated lifting platform with an initially dipping length of ~ 1 mm, as shown in Supplementary Movie 2. Under this condition, the dipping rate and the exfoliation rate kept a balance, and the GICP dipped in the electrolyte remained almost a constant length of ~ 1 mm. The products show a C/O atomic ratio of ~ 1.7 . These results suggest that ~ 1 -mm-long GICP was oxidized and exfoliated within 12 seconds. This gives another evidence that the EC oxidation process finished in a few seconds.

We have added the above data and related discussions in the revised manuscript.

4. The anodic electrocatalytic oxygen evolution reaction of water occurs under an applied voltage, and it contains four elementary reactions. More data are needed to support the existing of $*OH$, $*O$ and $*OOH$.

Reply: We thank the reviewer very much for kind suggestion.

It has been reported that 2,2,6,6-tetramethylpiperidin-1-oxyl (TEMPO) can efficiently suppress the formation of oxygen radicals from water electrolysis (*J. Am. Chem. Soc.*, 137, 13927, 2015). We studied the influence of addition of TEMPO on the

formation of EGO sheets during the optimized EC oxidation process of GICP (~50 wt. % H₂SO₄ aqueous solution, 5V). All the as-synthesized products were exfoliated in water to form suspensions with a concentration ~2 mg/mL. After standby 12 hours, the EGO products achieved without (sample I) and with the addition of a small amount (4‰, sample II) of TEMPO showed good stability with no obviously sediment, while the product achieved with the addition of a relatively large amount of TEMPO (10‰, sample III) totally precipitated. Elemental analyses show that adding 4‰ and 10‰ of TEMPO led to a decrease in C/O ratio of the products from ~1.8 to ~2.2 and ~2.6, respectively (see Supplementary Note 8). This gives strong evidence of the existing of oxygen radicals and their key role in the synthesis of GO during our EC oxidation process.

We have added the above data and the related discussions in the revised manuscript and Supplementary Note 8.

Reviewer #2 (Remarks to the Author):

This manuscript reports a fast and green method to prepare graphene oxide by electrochemical oxidation of flexible graphite paper in aqueous H₂SO₄ solution. This is an interesting work, and the results are important. I recommend accept this paper for publication after minor revisions.

Reply: We thank the reviewer very much for positive comments.

1. The electrochemical synthesis of graphene oxide has been studied by several groups. The typical papers are listed as follows. (a) Carbon, 2016, 100, 540-545. Synthesis of graphite oxide by electrochemical oxidation in aqueous perchloric acid; (b) J. Mater. Chem. A, 2014, 2, 15428-15436. Synthesis of graphene oxide nanosheets by electrochemical exfoliation of graphite in cetyltrimethylammonium bromide and its application for oxygen reduction; (c) Chem. Commun., 2016, 52, 5714. One-step electrochemical synthesis of nitrogen and sulfur co-doped, high-quality graphene oxide. The authors should compare their work with the previous papers to clarify the novelty and importance of this submission.

Reply: We thank the reviewer very much for kind suggestion. We have added the discussions to compare our work with those reported in the papers mentioned by the reviewer to further clarify the novelty of our method in the revised manuscript. The detailed comparison between different EC methods was also given in Supplementary Tables 2 and 3.

2. The authors described: “However, the present synthesis methods depend on the reactions of graphite with strong oxidizing mixtures containing one or more concentrated acids,……, which is 2 –4 orders of magnitude faster than the present methods”. However, the comparison of the new method with Hummers methods is based on different raw materials. The authors used flexible graphite paper (FGP) as raw material rather than natural graphite as used in Hummers method. Thus, the above comparison is unfair.

Reply: We thank the reviewer very much for the kind comment.

In order to rule out the influence of raw material on the oxidation rate, we have compared the synthesis of GO by using Hummers method with FGP and graphite powder as raw material under the same conditions. It was found that the oxidation degree of GO achieved from FGP (C/O ~2.8) is much lower than that from graphite powder (C/O ~2.1) after reaction for ~8 h, and the charging state of the former (zeta potential ~-17.2 mV) is also weaker than the latter (zeta potential ~-45.7 mV). This confirms that the fast oxidation of graphite in our EC method is indeed caused by the EC oxidation process rather than the raw material.

The above data and related discussions have been added in the revised manuscript and Supplementary Note 1.

3. The structure and properties of FGP should be studied more clearly. What are the differences between FGP and graphite rod or powders?

Reply: We thank the reviewer very much for kind suggestion.

The structure and properties of FGP were systematically studied by Raman

spectroscopy, XPS, TG-MS, and XRD. FGP is an industrial product of natural graphite, which is produced by roller-pressing of expanded graphite flakes (EGFs) without any binder. It is very flexible, and usually has a thickness from micrometers to millimeters, width up to meters, and length up to kilometers (Supplementary Figs. 3a,b). SEM measurements show that the EGFs are interlocked together in the FGP (Supplementary Figs. 3c-f). Moreover, FGP shows similar Raman, XPS, TG, MS spectra and XRD pattern with graphite (Supplementary Fig. 4), indicating their high purity and similar structure. As a result, FGP has good mechanical properties with tensile strength of 4 – 5 MPa, and excellent electrical conductivity comparable with HOPG. Furthermore, as shown in the manuscript, FGP has a good tolerance to the volume expansion caused by intercalation. It ensures good mechanical strength and electrical conductivity of GICP, which are the prerequisite for further EC oxidation reaction. Together with the good flexibility and scalable length, FGP is an ideal raw material for continuous EC synthesis of GO.

In our method, the EC intercalation and oxidation reaction must be driven by applied voltage. Therefore, graphite powders cannot be used directly because of the very small size and have to be fabricated into macroscopic electrode to ensure voltage supply. Graphite rods are a kind of macroscopic electrodes that are synthesized with fine graphite powders and binder by pressing and sintering at high temperature, but they are very brittle. We have tried to use graphite rod as raw material in our EC process, but they were broken into small pieces even after short time intercalation, and consequently could not be used for further oxidation.

The above results and discussions have been given as Supplementary Information in the revised manuscript

REVIEWERS' COMMENTS:

Reviewer #1 (Remarks to the Author):

All the comments were replied well. Accept as it is.

Reviewer #2 (Remarks to the Author):

This paper has been properly revised. Now it is acceptable for publication.